# Comparison of Canopy Closure Estimation of Plantations Using Parametric, Semi-Parametric, and Non-Parametric Models Based on GF-1 Remote Sensing Images

**Jiarui Li** [1] and **Xuegang Mao** [1,2,*]

1   School of Forestry, Northeast Forestry University, Harbin 150040, China; lijiarui_604@163.com
2   Key Laboratory of Sustainable Forest Ecosystem Management-Ministry of Education,
    Northeast Forestry University, Harbin 150040, China
*   Correspondence: maoxuegang@aliyun.com; Tel.: +86-1-820-460-9096

**Abstract:** Canopy closure (CC) is an important parameter in forest ecosystems and has diverse applications in a wide variety of fields. Canopy closure estimation models, using a combination of measured data and remote sensing data, can largely replace traditional survey methods for CC. However, it is difficult to estimate the forest CC based on high spatial resolution remote sensing images. This study used China Gaofen-1 satellite (GF-1) images, and selected China's north temperate Wangyedian Forest Farm (WYD) and subtropical Gaofeng Forest Farm (GF) as experimental areas. A parametric model (multiple linear regression (MLR)), non-parametric model (random forest (RF)), and semi-parametric model (generalized additive model (GAM)) were developed. The ability of the three models to estimate the CC of plantations based on high spatial resolution remote sensing GF-1 images and their performance in the two experimental areas was analyzed and compared. The results showed that the decision coefficient ($R^2$), root mean square error (RMSE), and relative root mean square error (rRMSE) values of the parametric model (MLR), semi-parametric model (GAM), and non-parametric model (RF) for the WYD forest ranged from 0.45 to 0.69, 0.0632 to 0.0953, and 9.98% to 15.05%, respectively, and in the GF forest the $R^2$, RMSE, and rRMSE values ranged from 0.40 to 0.59, 0.0967 to 0.1152, and 16.73% to 19.93%, respectively. The best model in the two study areas was the GAM and the worst was the RF. The accuracy of the three models established in the WYD was higher than that in the GF area. The RMSE and rRMSE values for the MLR, GAM, and RF established using high spatial resolution GF-1 remote sensing images in the two test areas were within the scope of existing studies, indicating the three CC estimation models achieved satisfactory results.

**Keywords:** canopy closure; multiple linear regression model; random forest; generalized addictive model; GF-1

## 1. Introduction

Canopy closure (CC), the percentage of land area covered by the vertical projection of tree crowns [1], is the most common variable estimated in forest inventories, because CC > 10% represents a criterion for the international definition of forests [2]. Leaf area index (LAI), defined as the total one-sided area of all leaves in the canopy within a defined region ($m^2/m^2$), mainly quantifies the amount of live green leaf material present in the canopy per unit ground area [3]. Both CC and Leaf area index(LAI) are commonly used to characterize the structure and function of forest ecosystems, but they refer to different aspects: CC always refers to subcompartment divisions, land classification, stand type classification, and stand quality evaluation, while LAI is an important structural parameter

related to the energy and mass exchange characteristics of terrestrial ecosystems such as photosynthesis, respiration, transpiration, carbon and nutrient cycle, and rainfall interception [3–7]. The main purpose of this study is to use remote sensing technology to replace the traditional method of field forest measurements, so CC is more appropriate and convenient for application.

The traditional methods of CC measurement include visual assessment, sample points, transects, canopy projections, observation tubes, and canopy instrument analysis [8]. These methods are manual field measurements, which need considerable human and material resources, and it is difficult to obtain CC estimates at a regional scale. Estimation of CC based on remote sensing was originally performed using aerial photos of the target area, but because the forest area is very large and the definition of the forest boundary will change over time, using aerial photos to obtain CC is too expensive [9]. Use of light detection and ranging (LiDAR) data can aid in the estimation of CC at a pixel level; however, it is not a cost-effective resource for long-term repeat measurements and cannot be used for historical analyses. Although LiDAR or integration of LiDAR with other sensor data in combination with field measurements is considered to be the best way to estimate CC, automation and extrapolation to a larger regional scale is still a challenge [10–14]. The estimation of CC based on microwave data is also limited because of issues with system availability, data processing, and heterogeneous interactions between bands and forest structure [15–18].

Optical satellite remote sensing images have the characteristics of effectiveness, reproducibility, and low cost, and are well-established processing methods. Therefore, a variety of optical remote sensing data sources have been successfully applied to the estimation of forest parameters such as CC. Hyperspectral and multi-angle data from optical remote sensing products are not the best remote sensing data sources for CC estimation due to complex data processing [19,20]. Medium- and low-spatial resolution optical remote sensing data (e.g., Moderate-resolution Imaging Spectroradiometer (MODIS)) cannot always meet forest canopy estimation requirements [21]. Although optical satellite imagery with medium spatial resolution is currently the main data source for CC estimation, it cannot detect small changes in closure at the forest stand scale [22–25]. Therefore, high spatial resolution remote sensing data provide a more effective data source for CC estimation with higher spatial and temporal resolution and lower acquisition costs. Studies have shown that high spatial resolution data can not only be used to verify the estimation of CC from lower resolution data, but can also reduce the estimation error [26–28].

The remote sensing inversion model of forest canopy density mainly includes physical and statistical models. The physical model is mainly represented by the geometrical optics model and radiative transfer model [29]. However, it is more complex and needs a large number of model parameters; beyond that, non-unique solutions and other problems also make it difficult for it to be popularized and applied widely [30]. The statistical model can be combined with remote sensing data and a small number of measured samples to predict the canopy closure of non-sampled areas. Compared with the physical model, the statistical model is an economical and efficient method to predict canopy closure. The remote sensing factors such as remote sensing band, vegetation index, and texture features, as well as soil characteristics, seasonal information, topography, and other auxiliary information are often used to estimate canopy closure by using statistical models [31–33]. With a variety of remote sensing information, vegetation indices show a very good performance when predicting canopy closure [34–37]. Statistical models are generally divided into parametric models, semi-parametric models, and non-parametric models. Thus, the performance of three kinds of models combined with vegetation index to estimate canopy closure needs to be further studied [9].

Earth observation satellites such as the China Gaofen-1 satellite (GF-1) have been successively introduced, greatly enriching the availability of high resolution multispectral remote sensing data. This data source has great potential in the estimation of CC and is worthy of further study. Therefore, in this study, high-resolution GF-1 remote sensing images were used. The northern temperate forest (Wangyedian Forest Farm, Chifeng City, Inner Mongolia Autonomous Region, China) and subtropical forest (Gaofeng Forest Farm, Gaofeng City, Guangxi Zhuang Autonomous Region, China) were

selected as experimental areas. We developed a parametric model (multiple linear regression (MLR)), semi-parametric model (generalized additive model (GAM)) and non-parametric model (random forest (RF)) for estimating CC with GF-1 high spatial resolution images, and compared the performance of the three models in plantation forests. The study had three main objectives: to evaluate the performance of GF-1 high spatial resolution remote sensing imagery to estimate CC; to identify which of the three types of model was the best method; and to find whether the performance of the three models was consistent in the two experimental areas.

## 2. Materials and Methods

### 2.1. Study Area

Wangyedian Forest Farm (WYD) is located in the southwest of Harqin Banner, Chifeng City, Inner Mongolia Autonomous Region, China, at the intersection of the Inner Mongolia, Hebei, and Liaoning provinces. Its geographical location is 118°09′–118°30′ E, 41°35′–41°50′ N (Figure 1). The forest farm has a moderate-temperate continental monsoon climate with an altitude of 800–1890 m. The relative height of the mountains is generally 200–400 m, and the slope is 15–35°. The area of the forest farm is $2.47 \times 10^4$ hm$^2$ (1 hm$^2$ = $1 \times 10^4$ m$^2$). It was created in 1956 and is currently a pilot of the Asia-Pacific Forest Restoration and Sustainable Management Organization (APPNet) multifunctional forestry demonstration base and a national key forest fine seed base. The forest coverage rate is 92.10%, and the forest area covers $2.33 \times 10^4$ hm$^2$, of which the plantation is $1.16 \times 10^4$ hm$^2$. The tree species of the plantation are mainly oil pine (*Pinus tabuliformis* Carrière) and larch (*Larix principis-rupprechtii* May), of which 51% of the planted area is oil pine and 47% is larch. Gaofeng Forest Farm (GF) is located in Nanning City, Guangxi Zhuang Autonomous Region, China. It is belt-shaped and surrounds the northern part of Nanning City. The geographical location is 108°08′–108°53′ E, 22°49′–23°15′ N (Figure 1). The relative height of the forest farm is generally 150–400 m and the slope is 20–30°. It is located in the humid subtropical monsoon climate area, with an average annual temperature of 21.6 °C and an average annual rainfall of 1300 mm. GF was created in 1953, and the total area of the forest land under management was $8.70 \times 10^4$ hm$^2$. Non-forest land covered $1.21 \times 10^2$ hm$^2$, and forest coverage was 87.50%. In 1998, the rapid development of fast-growing eucalyptus and cedar short-cycle industrial raw material forests began. The forest farm developed a rapid-growing forest area of $4.7 \times 10^4$ hm$^2$, of which eucalyptus covers $2.50 \times 10^4$ hm$^2$. The reason why WYD and GF were chosen as study areas is because they both represent typical plantations in northern and southern China, and the spatial distribution of forests is obvious.

### 2.2. Data

#### 2.2.1. Remote Sensing Data

Launched on 26 April 2013, GF-1 was the first satellite successfully launched by the China High-resolution Earth Observation System (CHEOS) and the first of China's high-resolution Earth observation system satellites. The GF-1 satellite is fitted with two panchromatic and multispectral (PMS) cameras which can provide pan and multispectral data with nadir resolutions of 2 m and 8 m, respectively, across an imaging swath of 60 km. Radiometric resolution of GF-1 is 10 bit. The GF-1 multispectral remote sensing images include the blue band (0.45 μm–0.52 μm), green band (0.52 μm–0.59 μm), red band (0.63 μm–0.69 μm), and near infrared band (0.77 μm–0.89 μm). The GF-1 satellite images were downloaded through the Remote Sensing Market Service Platform of the Chinese Academy of Sciences [38]. The WYD study area is covered by four scenes and the GF study area by two scenes (specific image information is shown in Table 1). The GF-1 remote sensing images were sequentially processed by radiation calibration, atmospheric correction, and terrain correction. In the preprocessing of all images, except for the parameters of the image, the order and settings of the preprocessing operations were the same. This study was based on the radiation calibration coefficients

of GF-1 remote sensing images published by the China Resources Satellite Application Center [39]. The GF-1 images were subjected to atmospheric correction using the dark method [40]. The terrain in the WYD and GF test areas was relatively large, and the GF-1 image was affected by the terrain. This study used the C correction method which has been widely used for terrain correction [41–43].

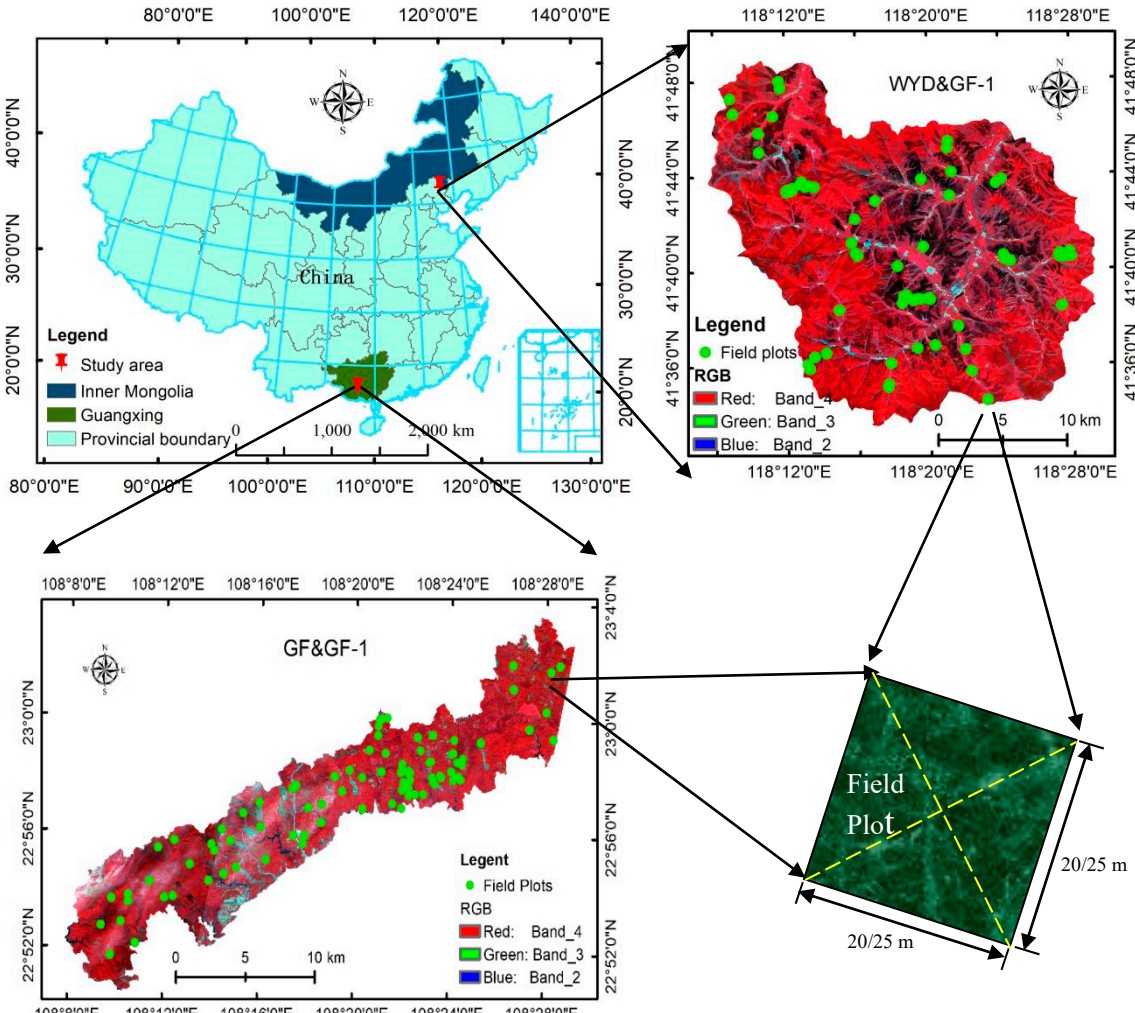

**Figure 1.** Schematic diagram of study area location, data source, and plot distribution. WYD— Wangyedian Forest Farm, GF— Gaofeng Forest Farm, GF-1—China Gaofen-1 satellite.

**Table 1.** Parameter information for Gaofen-1 (GF-1) remote sensing images in the two study areas.

| Research Area | Scenery Serial Number | Imaging Time | Solar Elevation Angle (°) | Solar Azimuth (°) | Cloud Cover (%) |
|---|---|---|---|---|---|
| Wangyedian (WYD) | 3858265 | 8 July 2017 | 69.423 | 155.338 | 0 |
| | 3858264 | 8 July 2017 | 69.191 | 155.798 | 1 |
| | 3857940 | 8 July 2017 | 69.226 | 154.376 | 0 |
| | 3857939 | 8 July 2017 | 68.997 | 154.839 | 3 |
| Gaofeng (GF) | 3255633 | 22 January 2017 | 45.450 | 161.971 | 3 |
| | 3255824 | 22 January 2017 | 45.600 | 162.360 | 0 |

### 2.2.2. Field Data

The measurement dates of CC in the two study areas of WYD and GF were 20–30 September 2017 and 14–25 January 2018. The plot setting and measurement methods were the same. The plot was set in a patch of the same forest type that was much larger than the size of the plot to minimize edge

effects and reduce the impact of image correction errors. The Global Positioning System (Zenith45) was used to measure the coordinates of the four corners of each square plot, and calculated the center position of the plot. The positioning accuracy of the plot reached the sub-meter level. The starting diameter of each plot was 5 cm. The forest parameters of each tree for each plot were measured (tree species, diameter at breast height, tree height, crown width, CC) and the information for each plot was calculated according to the tree species (Table 2). The transect method was used to measure CC, that is, setting a sample line along two diagonal lines and recording the length of the vertical projection of the canopy on the sample line along each transect. The total length of the vertical projection of the sample line crown was divided by the length of the diagonal to obtain the degree of depression on the diagonal. The average value of CC on the two diagonal lines was the CC value of the plot. This method is the most reliable way to estimate the vertical CC of forests and can be directly compared with the CC estimated by remote sensing data [8]. The plots in the two test areas of WYD and GF were 25 m × 25 m and 20 m × 20 m. The plots in the GF test area were set to 20 m × 20 m, mainly due to the major stand density (strains/hm$^2$), which reduced the field investigation workload. Both the WYD and GF test areas had 80 plots which were evenly distributed, covering the two major plantation species in the regions (Figure 1).

**Table 2.** Statistics of canopy closure (CC) plots in the two study areas.

| Research Area | Stand Type | Number of Plots | Elevation (m) | Plant Number Density (Plants/hm$^2$) | Thoracic High Sectional Area (m$^2$/hm$^2$) | Accumulation (m$^3$) | Canopy Closure |
|---|---|---|---|---|---|---|---|
| Wangyedian (WYD) | Pine | 40 | 981–1370 | 528–2816 | 0.13–1.43 | 5.77–22.58 | 0.22–0.86 |
| | Larch | 40 | 1074–1355 | 256–5264 | 0.07–2.81 | 5.39–25.35 | 0.39–0.82 |
| Gaofeng (GF) | Eucalyptus | 60 | 113–374 | 900–3450 | 0.54–0.69 | 1.29–13.33 | 0.35–0.92 |
| | Chinese fir | 20 | 142–181 | 650–1400 | 0.49–0.83 | 6.47–12.36 | 0.63–0.82 |

*2.3. Methods*

2.3.1. Remote Sensing Variable Extraction

In addition to the green band (GREEN) and near-infrared (NIR) reflectance of GF-1, the remote sensing variable also used eight vegetation indices as the independent variables (Table 3). When the degree of CC reaches a certain level, the spectral signal no longer increases with the increase of CC, resulting in the phenomenon of "saturation" of the spectrum [44]. Therefore, it was necessary to find suitable vegetation index variables to avoid the problem of spectral saturation impacting closure estimates.

**Table 3.** Vegetation indexes and calculation formula.

| Variables Name | Variables | $R^2$ (WYD) | $R^2$ (GF) | Calculation Formula | References |
|---|---|---|---|---|---|
| Green band | GREEN | 0.43 | 0.36 | - | - |
| Near-infrared band | NIR | 0.57 ** | 0.52 ** | - | - |
| Difference Vegetation Index | DVI | 0.39 * | 0.39 ** | $DVI = NIR{-}R$ | Bannari et al. (1995) [45] |
| Ratio Vegetation Index | RVI | 0.49 ** | 0.41 | $RVI = NIR/R$ | Colombo et al. (2003) [46] |
| Simple Ratio Index | SR | 0.42 * | 0.36 | $SR = R/NIR$ | Colombo et al. (2003) [46] |
| Normalized Difference Vegetation Index | NDVI | 0.39 | 0.43 * | $NDVI = (NIR{-}R)/(NIR + R)$ | Yue et al. (2007) [47] |
| Return to Vegetation Index | RDVI | 0.51 ** | 0.49 * | $RDVI = \sqrt{NDVI \times DVI}$ | Roujean et al. (1995) [48] |

**Table 3.** *Cont.*

| Variables Name | Variables | $R^2$ (WYD) | $R^2$ (GF) | Calculation Formula | References |
|---|---|---|---|---|---|
| Perpendicular Vegetation Index | PVI | 0.39 | 0.37 | *PVI = 0.939× NIR−0.344× R + 0.09* | Richardson et al. (1977) [49] |
| Soil Adjustment Vegetation Index | SAVI | 0.45 * | 0.46 * | *SAVI = (NIR−R)/(NIR + R + L) × (1 + L)* | Huete (1988) [50] |
| Modified Soil Adjustment Vegetation Index | MSAVI | 0.40 | 0.42 * | $MSAVI = [(2 \times NIR + 1) - \sqrt{(2 \times NIR)^2 - 8 \times (NIR - R)}]/2$ | Qi et al. (1994) [51] |

Note: NIR and R are the near-infrared and red surface reflectance; L is the soil adjustment coefficient. The value here is 0.5; **: Significant correlation at the 0.01 level; *: Significant correlation at the 0.05 level.

### 2.3.2. Canopy Closure Estimation Models

Multiple Linear Regression

The main objective of regression analysis research is the statistical relationship between the variables. It is based on a large number of experiments and observations of objects, and is used to find the statistical rules hidden in seemingly uncertain phenomena. The analysis can not only reveal the influence of independent variables on the dependent variable, but can also control and predict using regression equations. Multiple linear regression has a wide range of applications. In this study, the significant independent variables were selected through the stepwise screening method in stepwise regression, eliminating multicollinearity between independent variables (Equation (1)).

$$y = b_0 + \sum_{i=1}^{n} b_i x_i \tag{1}$$

where, $b_0$ is a constant term, that is, the intercept; $b_i (i = 1, 2, \ldots, n)$ is the partial regression coefficient.

Generalized Additive Model (GAM)

The GAM is based on a linear combination of piecewise polynomials. These polynomials, describing the contribution of each independent variable, are combined into an independent function (Equation (2)) defined by a smoothing spline [32]. The dimension (k) is an important parameter in GAM, which is used to fit the smoothing function. To avoid the occurrence of overfitting, the value of k was 3 [23]. The connection function used the default identity function of GAM after importing the data. The modeling algorithm was the default constraint maximum likelihood. In the multivariate model, although each additional variable can reduce the root mean square error (RMSE) of the fitted model, it also increases the risk of overfitting, so three independent variable selection principles were followed in GAM modeling. Firstly, there were three independent variables. When the number of independent variables was 3–4, the RMSE of the model was relatively stable. The calculation load was greater when the number was greater than four. Secondly, when the independent variables were introduced and CC was modeled separately, the GAM was significant at the level of 0.05 ($p < 0.05$), Thirdly, the three variables that had the greatest effect on the degree of depression were retained using the method of elimination. First, all variables were introduced, and each was eliminated in turn. The RMSE was calculated and the variable with the largest RMSE was removed. The remaining variables were sequentially eliminated using this process until there were three variables left.

$$y = b_0 - \sum_{i=1}^{n} f_i(x_i) \tag{2}$$

where, $b_0$ is a constant term, that is, the intercept; $f_i(x_i)(i = 1, 2, \ldots, n)$ is a smoothing function for each independent variable.

Random Forest

Many studies have used remote sensing data and regression trees to calculate the relevant attribute values of forests [15,52]. Random forest is a special implementation of regression trees. In this algorithm, users can set, at the nodes of each tree, the number of trees to be established and the number of independent variables to be introduced [53]. Each execution of RF is an introductory aggregation of all trees. In the process of regression, the RF method selects some samples to hold back from random sampling, so some of the samples are not used to build the regression tree. The average of the prediction results of all trees is the final prediction result. The two important parameters of the modeling are ntree and mtry. Ntree is the number of times of bootstrap resampling. In general, the regression error will stabilize after the number of regression trees reaches 500. To ensure the reliability of the results, the ntree parameter value was set to 1000 in this study. Mtry is the number of independent variables introduced randomly each time the regression tree is established. To make the model fit the best, an RF model was developed for 10 cases, with the number of independent variables introduced ranging from 1 to 10 when using the three data types for modeling. The mean values of the squared residuals obtained from each case were compared, and the number of independent variables that corresponded to the minimum mean squared residual represented the parameter value of mtry. The mtry of GF-1 data was 1.

The importance of variables can be evaluated by using the RF model. The basic concept is to add noise to a related feature (that is, a feature that may play an important role in prediction accuracy). If the prediction accuracy of RF model decreases, the variable is important; the greater the degree of accuracy reduction (mean decrease in MSE), the more important the feature variable. In this study, the model under the optimal parameter setting was trained for 100 times to obtain the average importance of each variable, and the importance of each variable was evaluated according to the changing degree of prediction accuracy of the model. The above three models were implemented in the statistical software R, RF in the random Forest package and GAM in the mgcv package in R.

*2.4. Model Inspection*

The decision coefficient ($R^2$), root mean square error (RMSE), and relative root mean square error (rRMSE) were selected to evaluate the accuracy of the models. $R^2$ is used to test the goodness of fit of the regression line, reflecting the correlation between the independent variable and the dependent variable: the larger the $R^2$, the higher the goodness of fit, and the better the correlation (Equation (3)). The root mean square error refers to the RMSE of the measured and predicted values, affected by the magnitude of the object to be evaluated (Equation (4)). The rRMSE is the root mean square of relative error, which is independent of the magnitude of the object to be evaluated. It can eliminate the impact of the dimension and is used to compare models with large differences in the overall range (Equation (5)). The model's prediction effect improves alongside a decrease in the RMSE and rRMSE. From the two experimental areas, 70% of data points were randomly selected as modeling samples, and the remaining 30% were used as test samples to calculate $R^2$, RMSE, and rRMSE.

$$R^2 = 1 - \frac{\sum_{i=1}^{n} (y_i - \hat{x}_i)^2}{\sum_{i=1}^{n} (y_i - \overline{x})^2} \tag{3}$$

$$RMSE = \sqrt{\frac{1}{(n-1-p)} \sum_{i=1}^{n} (y_i - \hat{x}_i)^2} \tag{4}$$

$$rRMSE = \frac{RMSE}{\overline{x}} \tag{5}$$

where *p* is the number of variables, $R^2$ is the determination coefficient, $Y_i$ is the sample values for the sample *i*, $\bar{x}$ is the average of the sample, $\hat{x}_i$ is the estimated value of the corresponding sample *i* (*i* = 1, 2, ..., *n*), and *n* is the number of samples.

## 3. Results

The $R^2$, RMSE and rRMSE ranges for the MLR, GAM, and RF models of WYD were 0.45–0.76, 0.0632–0.0953 and 9.98–15.05%, respectively (Table 4). Among the three models, the semi-parametric model (GAM) performed best, and the parametric model (MLR) performed the worst. The RMSE (rRMSE) values of the GAM model were reduced by 0.0211 (3.33%) and 0.0321 (5.05%), respectively, compared with the other models. The value of $R^2$, RMSE, and rRMSE of the MLR, GAM, and RF established in the Gaofeng area ranged from 0.40 to 0.59, 0.0967 to 0.1152, and 16.73% to 19.93% respectively. Among the three models, the GAM performed best, and the non-parametric model (RF) performed the worst. The GAM reduced the RMSE (rRMSE) compared with the parametric model (MLR) and non-parametric model (RF) by 0.0051 (0.88%) and 0.0185 (3.20%), respectively.

**Table 4.** Three models and tests established in the two study areas.

| Test Area | Model | Model Form | $R^2$ | *RMSE* | *rRMSE* |
|---|---|---|---|---|---|
| WYD (Wangyedian) | *MLR* | CC = −96.55 × DVI + 6.79 × NIR + 240.73 × RDVI − 0.37 × RVI−106.85 × SAVI + 47.37 × SR − 37.85 | 0.69 | 0.0843 | 13.31% |
| | *GAM* | CC = f(RDVI) + f(RVI) + f(NIR) + 0.60 | 0.76 | 0.0632 | 9.98% |
| | *RF* | CC = f(GREEN, NIR, DVI, RVI, SR, NDVI, RDVI, PVI, SAVI, MSAVI) | 0.45 | 0.0953 | 15.05% |
| GF (Gaofeng) | *MLR* | CC = −3666.73 × DVI + 34.00 × MSAVI − 1813.71 × NDVI + 22.25× NIR + 10430.38 × RDVI − 5020.57 × SAVI − 1.03 | 0.48 | 0.1018 | 17.61% |
| | *GAM* | CC = f(DVI) + f(PVI) + f(NIR) + 0.58 | 0.59 | 0.0967 | 16.73% |
| | *RF* | CC = f(GREEN, NIR, DVI, RVI, SR, NDVI, RDVI, PVI, SAVI, MSAVI) | 0.40 | 0.1152 | 19.93% |

MLR—Multiple Linear Regression, GAM—Generalized Additive Model, RF—Random Forest, CC—Canopy Closure, DVI—Difference Vegetation Index, NIR—Near-infrared Band, RDVI—Return To Vegetation Index, RVI—Perpendicular Vegetation Index, SAVI—Soil Adjustment Vegetation Index, SR—Simple Ratio Index, GREEN—Green Band, PVI—Perpendicular Vegetation Index, MSAVI—Modified Soil Adjustment Vegetation Index.

The number of independent variables to establish the parametric model (MLR) screened by stepwise regression was six for each experimental area and there were four identical variables (Difference Vegetation Index (DVI), near-infrared band (NIR), Return to Vegetation Index (RDVI), and Soil Adjustment Vegetation Index (SAVI)). Only one of the three independent variables of the GAM was the same, NIR. The RF model used all 10 variables (Green band (GREEN), NIR, DVI, Ratio Vegetation Index (RVI), Simple Ratio Index (SR), Normalized Difference Vegetation Index (NDVI), RDVI, Perpendicular Vegetation Index (PVI), SAVI, and Improved Soil Adjustment Vegetation Index (MSAVI)) (Table 4). The performance of the MLR, GAM, and RF established in the two study areas was consistent, that is, the GAM had the highest modeling accuracy, and the MLR was second, and RF had the lowest accuracy. From the comparison of the two study areas, the accuracy of the three models established in WYD was higher than that in GF. The reason was that part of the GF-1 remote sensing image in GF was covered by thin clouds (Figure 1).

The scatter plots of the estimated CC values compared with the actual measured values of the parametric model (MLR), semi-parametric model (GAM), and parametric model (RF) established in the two test areas are shown in Figure 2. We can see from the positional relationship of the y = x line that distribution of the scatter plots of MLR and GAM in the two test areas had a large degree of similarity, but the GAM model was better than MLR with estimation accuracy. The errors of the MLR and GAM models were mainly overestimation of high depression closure values and underestimation of low depression closure values.

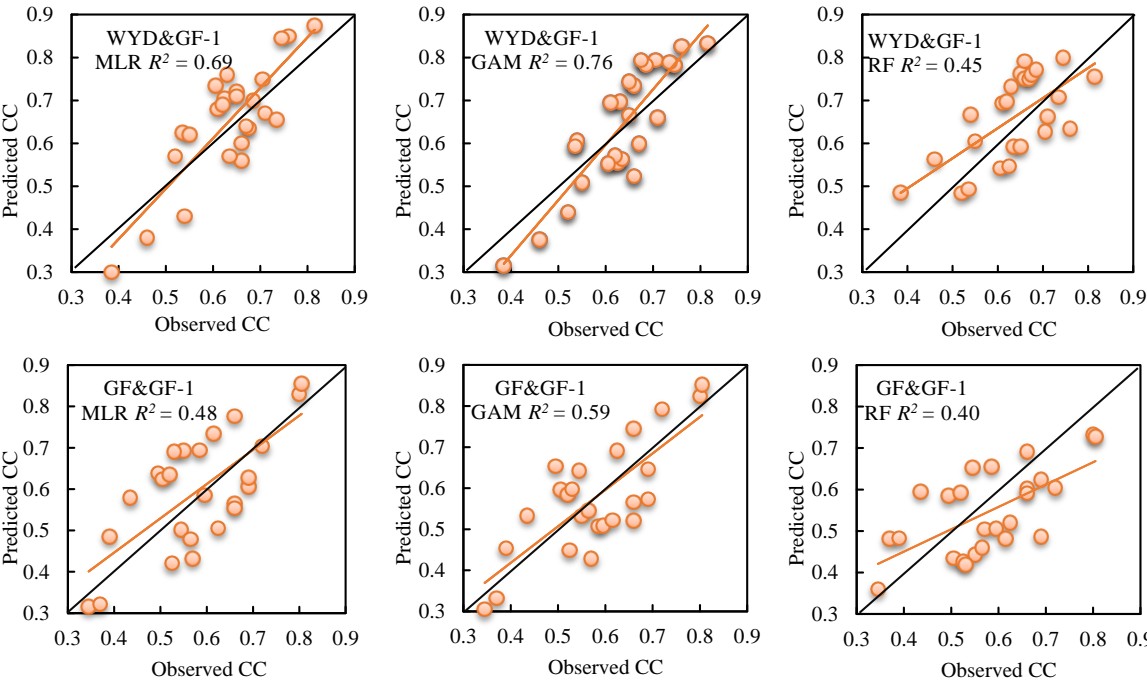

**Figure 2.** Scatter plot of measured values of canopy closure (CC) and predicted values of the optimal model. Note: The black solid line in the figure is the y = x straight line, and the orange straight line is the linear regression fitting line between the measured and predicted values. WYD—Wangyedian Forest Farm, GF—Gaofeng Forest Farm, GF-1—China Gaofen-1 satellite, RF—Random Forest, MLR—Multiple Linear Regression.

The average importance of the predictor of RF model in the two experimental areas is shown in Figure 3. The importance of variables for both WYD and GF were similar. RDVI and NDVI were the most important variables, followed by RVI and SR, while NIR and GREEN had the lowest mean decrease in MSE. According to the analysis of the importance of variables in the two experimental areas, the importance of vegetation index was higher than that of band reflectance data (NIR and GREEN). The importance of normalized vegetation index variable (RDVI, NDVI) was higher than that of other vegetation index variables. This revealed a strong relationship between RDVI and NDVI and the predicted variable (canopy density). Importance variables were concentrated in several certain variables, indicating that other predictors were noise variables.

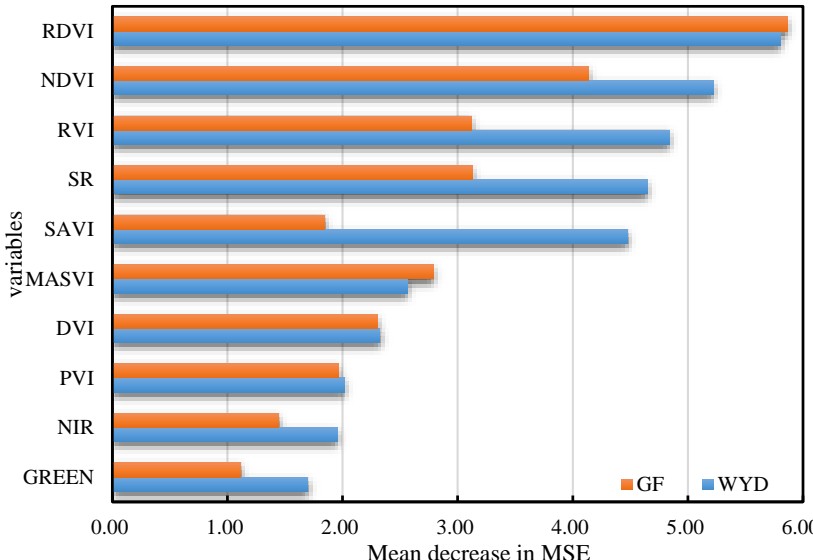

**Figure 3.** Variable importance for the random forest (RF) models of the two pilot regions. RDVI—Return To Vegetation Index, NDVI— Normalized Difference Vegetation Index, RVI—Perpendicular Vegetation Index, SR—Simple Ratio Index, SAVI—Soil Adjustment Vegetation Index, MSAVI—Modified Soil Adjustment Vegetation Index, DVI—Difference Vegetation Index, PVI—Perpendicular Vegetation Index, NIR—Near-infrared Band, GREEN—Green Band.

## 4. Discussion

Medium-resolution remote sensing images are currently the main source of remote sensing data used to estimate CC. The main reasons are that these data sources can be obtained free of charge and the time series is long [9,13]. For example, the Landsat series of satellites have been providing stable data for 45 years [54]. This study established a parametric model (MLR), a semi-parametric model (GAM), and a non-parametric model (RF) for two experimental areas based on GF-1 remote sensing images, and the ranges of RMSE and rRMSE were: 0.0632–0.1152 and 9.98–19.93%. The range of RMSE of the existing estimation model of CC was 0.07–0.13, and the rRMSE was about 20%. In this study, GF-1 remote sensing imagery was used to estimate CC. The accuracy is comparable to other remote sensing images, and in some cases, it was higher than the accuracy of other remote sensing images (such as Sentinel-2A MSI and Landsat 8 OLI) (Table 5).

**Table 5.** Comparison of GF-1 estimation of canopy closure (CC) and other sensors.

| Data Source | $R^2$ | RMSE | rRMSE% | Method | n | Reference |
|---|---|---|---|---|---|---|
| Aerial Multispectral Sensor | 0.79 | | | Parametric model (MLR) | 6 | Le´vesque (2003) [55] |
| SPOT5 | 0.68 | 0.06 | | Parametric model (partial least squares) | 39 | Wolter et al. (2009) [31] |
| SPOT5 | 0.52 | 0.05 | | Parametric model (partial least squares) | 40 | Wolter et al. (2009) [31] |
| Sentinel-2A MSI | 0.69 | 0.11 | 16.0 | Semiparametric model (GAM) | 19 | Korhonen et al. (2017) [13] |
| Landsat 8 OLI | 0.70 | 0.10 | 15.0 | Semiparametric model (GAM) | 19 | Korhonen et al. (2017) [13] |
| Landsat 8 OLI | | 0.12 | 12.0 | Nonparametric model (RF) | 60 | Halperin et al. (2016) [9] |
| RapidEye | | 0.12 | 12.3 | Nonparametric model (RF) | 60 | Halperin et al. (2016) [9] |

**Table 5.** *Cont*.

| Data Source | $R^2$ | RMSE | rRMSE% | Method | n | Reference |
|---|---|---|---|---|---|---|
| Landsat+LiDAR | 0.66 | 0.07 | | Nonparametric model (RF) | >100 | Ahmed et al. (2015) [33] |
| Landsat 8 OLI | | 0.13 | 14.2 | Nonparametric model (KNN) | 60 | Halperin et al. (2016) [9] |
| RapidEye | | 0.11 | 14.6 | Nonparametric model (KNN) | 60 | Halperin et al. (2016) [9] |

SPOT—Systeme Probatoire d'Observation de la Terre, MSI—Multispectral Imager, OLI—Operational Land Imager, LiDAR—Light Detection And Ranging.

The parametric model (MLR), semi-parametric model (GAM), and non-parametric model (RF) for the two study areas had significant differences in $R^2$. The semi-parametric model (GAM) had a relatively good fit because GAM can be used to fit variables with complex relationships, and the independent and dependent variables do not need to satisfy any hypothetical relationships and distributions. The smoothing spline had the ability to adjust the curve fit according to the data, regardless of whether the relationship between the data was linear or non-linear, so GAM was flexible and versatile [23,56]. The parametric model (MLR) showed the second best fit. When the relationship between the dependent variable and the independent variable is obviously linear, a linear regression model can be used to reflect the linear relationship between variables. In this study, there were only 56 samples, and the linear relationship between the independent and dependent variables was not obvious, but having multiple independent variables (six), can make up for the shortcomings of a weak linear relationship.

The scatter plots of MLR and GAM had similar distributions. This was because two of the three modeling variables for GAM were also modeling variables for MLR. The parametric model (MLR) and the semi-parametric model (GAM) used a univariate linear function and a smoothing spline, respectively, to fit the independent variables and the dependent variable, and both models calculated the intercept, and the models were an additional form between multiple variables. The three evaluation indicators of RF were all worse than MLR and GAM because of the number of modeling samples. Although non-parametric models do not require data to satisfy theoretical assumptions and the method is relatively easy to implement, small samples will limit the modeling effect and accuracy of RF, and will weaken the predictive ability to some extent [9,56,57]. Although the small sample number was not suitable for RF establishment, its RMSE and rRMSE were very good. Hence, the reliability of RF was high, but it was not the best model for the estimation of CC in this study. The number of samples is one of the factors affecting the accuracy of modeling.

In this study, the importance of eight vegetation indices predicted by RF model showed that two normalized vegetation indices (RDVI and NDVI) were higher than the other vegetation indices. For the normalized vegetation index could eliminate the effects of solar height angle, satellite observation angle, topographic change, cloud/shadow, and atmospheric attenuation, and meanwhile reflect the influence of vegetation canopy background. NDVI was a good predictor of canopy cover of arid forests in Africa [58]. Martin et al. (2015) also obtained similar conclusions when using stochastic forest model to estimate canopy density based on the vegetation index extracted by Landsat-8 [39]. SAVI and MSAVI were also reported to play an important role in the prediction of canopy density in some studies [9]. However, they were not obvious in our study, mainly because the forests in the two experimental areas were relatively dense and were not affected by the soil background.

The residuals predicted by the three models were all within a reasonable range in the two experimental areas (Figure 4). However, MLR and GAM models both had different degrees of overestimation and underestimation for low canopy density and high canopy density, respectively. The phenomenon is mainly because the change of vegetation index sensitivity to vegetation coverage. In areas with higher vegetation cover, the vegetation index tended to be compressed; in the area with lower vegetation cover, the vegetation index was exaggerated. Although there was no obvious

overestimation and underestimation for low canopy density and high canopy density in RF, the prediction of low canopy density and high canopy density by RF model was not ideal. The main reason was that the final prediction of the RF model was based on the average of each single tree generated by the bootstrap sample. Reference data sets contained fewer higher and lower canopy densities samples, so RF might underrepresent the tree structure and the prediction of RF tended towards averages. In other research, RF was reported to be the optimal model when using a large population of samples [33]. In general, GAM is relatively advantageous compared with MLR and RF in this study.

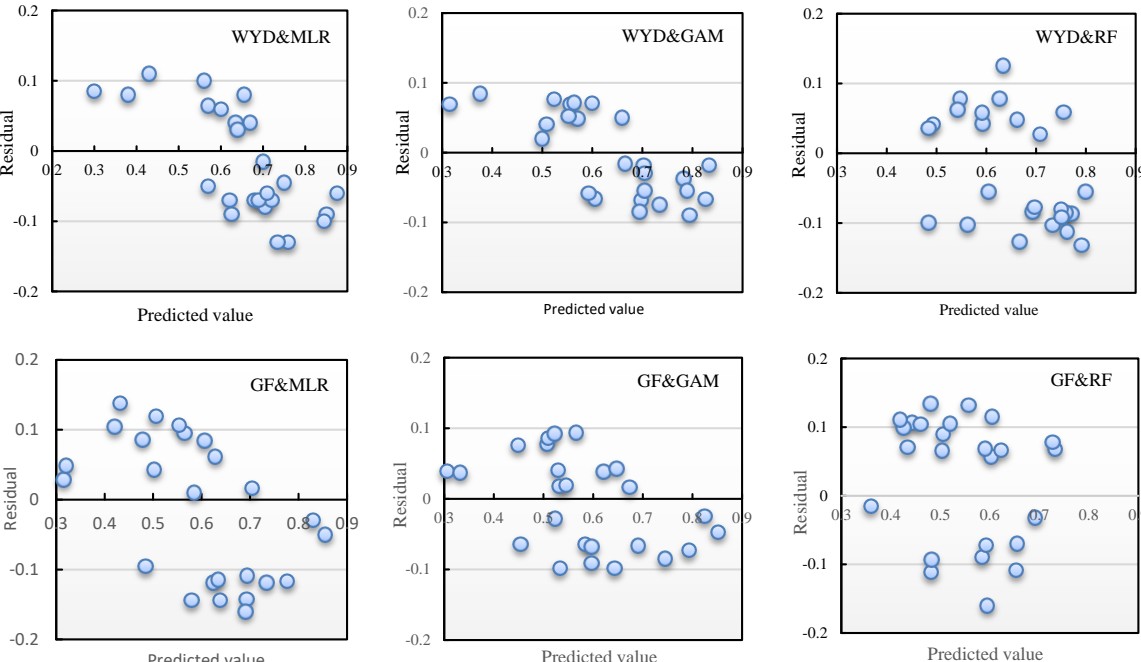

**Figure 4.** Residual diagram of canopy closure of two test areas estimated by three models. WYD—Wangyedian Forest Farm, MLR—Multiple Linear Regression, GAM—Generalized Additive Model, RF—Random Forest, GF—Gaofeng Forest Farm.

## 5. Conclusions

Canopy closure is a key parameter of forests, and plays an important role in forest management, investigation, and planning. It is also widely used in various fields related to ecology. A variety of modeling methods are effective for estimating CC. The focus of this research was to study the estimation of CC using a parametric model (MLR), a semi-parametric model (GAM), and a non-parametric model (RF) based on high spatial resolution remote sensing images (GF-1). The main conclusions drawn from the study are as follows. Firstly, establishing the three models using high spatial resolution remote sensing imagery (GF-1) to estimate the CC of the artificial forest can achieve satisfactory results. Secondly, the semi-parametric model (GAM) was better, showed strong generalization ability, and was more stable compared with the parametric model (MLR) and non-parametric model (RF). Thirdly, MLR, GAM, and RF are typical models of parametric, semi-parametric, and non-parametric models. However, the approach needs to be further tested with other models using high spatial resolution remote sensing images to estimate CC performance. This will make the conclusions of this study more universal. Fourthly, this study could be a data source and model for remote sensing in CC estimation. This study can provide a reference for the selection of data sources and model forms of remote sensing in canopy estimation and provide theoretical and technical support for transregional applications.

**Author Contributions:** J.L. contributed to the design of the study, conducted the fieldwork, performed the image processing and analysis, and prepared the manuscript. X.M. advised on the study design, conducted the fieldwork, image processing and analysis, contributed to manuscript writing and revision. All the authors have read and

approved the final manuscript. All authors provided editorial advice and participated in the review process. All authors have read and agreed to the published version of the manuscript.

**Funding:** National Key R&D Program of China: 2017YFD0600902. Fundamental Research Funds for the Central Universities: 2572019CP12, 2572018BA02.

**Acknowledgments:** We thank Leonie Seabrook, from Liwen Bianji, Edanz Group China (www.liwenbianji.cn/ac), for editing the English text of a draft of this manuscript.

**Conflicts of Interest:** The authors declare no conflict of interest.

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
