# Peer review of "Comparison of Canopy Closure Estimation of Plantations Using Parametric, Semi-Parametric, and Non-Parametric Models Based on GF-1 Remote Sensing Images"

_forests, doi:10.3390/f11050597_

Round 1
Reviewer 1 Report
GENERAL COMMENTS
Authors developed an enough evaluation of the use the multispectral sensors GF-1 for estimating canopy closure in temporal and subtropical forest of China. The objectives and research questions of the manuscript are important for the use of vegetation indices estimated from multispectral sensors in the measuring of canopy closure. Thus, research questions are well justified. Additionally, the structure of the article agrees with the research questions and match well to Forest criterions to publish. However, I found some issues in the methodology that authors can correct with not troubles. Thus, authors must correct these issues before publishing. These methodological problems are related to the use of the Random Forest algorithm in the analyses. I found also that authors must clarified the concept of Canopy closure and LAI (Leaf Area Index) in the introduction. Authors should also add the importance of the predictors for the three regression algorithms in both study areas. The importance of the predictors will allow to other researcher a better understanding of authors results.
INTRODUCTION
How the concept of Canopy closure (CC) is related to the concept of LAI (Leaf Area Index); both concepts seem high similar (LAI is defined as the one-sided green leaf area per unit ground surface area - LAI = leaf area / ground area, m2 / m2). Authors need to write in the introduction (or clarify differences) on the use of CC and no-use of LAI since many papers in remote sensing use both concepts.
I recommend this special issue in Remote Sensing: https://www.mdpi.com/journal/remotesensing/special_issues/leaf_area_index
METHODOLOGY
LINES -119 - 132 Authors need to add information on the nominal spatial resolution of GF-1, maybe information on radiometric resolution of GF-1, maybe cover of Gf-1 imagery, and information on download imagery of GF-1. The previous information is important because the readers of Forest are familiar with other multispectral global missions such as Landsat and Sentinel-2, but we do not so much on GF-1 mission.
Lines 202-221 Authors have some inexactitudes when they described the Random Forest algorithm. Authors must correct the Random Forest description.
First inexactitude (lines 207-210): Authors wrote that Random Forest makes an internal cross validation using the no-sampled data (literature refers as OOB – Out of Bag). Random Forest does not do a cross validation, Random forest makes a validation but do not make a cross validation. Authors also wrote that the use of OOB to validate the results of Random forest reduces the overfitting. It does not have sense; OOB and overfitting are not related.
Lines 214-215 Authors wrote that Mtry is the number of independent variables introduced randomly each time the regression tree is established; this is another inexactitude. Mtry is Number of variables randomly sampled as candidates at each split. It means that during the construction of each tree of the forest, the data is split (divided) using a number of variables equal to Mtry. Different variables are used in each split; for that reason, Random Forest is an algorithm that tends to reduce collinearity and, in some way, the overfit. However, Random forest is still affected by overfitting.
Lines 215-218 Authors wrote: To make the model fit the best, an RF model was developed for 10 cases, with the number of independent variables introduced ranged from 1 to 10 when using the three data types for modeling. Authors must describe with more accuracy what they did exactly. Maybe they did a Feature Selection Using Random Forest (see the link below), but I am not sure. (https://chrisalbon.com/machine_learning/trees_and_forests/feature_selection_using_random_forest/?fbclid=IwAR3r8hekOfA-6ajI3chF5gYnbK813tEli3JEJWRnWl-DWb5tx_zmA0cDqNc)
RESULTS
I recommend evaluating the construction of the Random forest models (including my previous comments of the methodology). Maybe the results could change.
I highly recommend the inclusion of the predictors importance for the three regression algorithms in both study areas. Also, authors must discuss this result in the discussion section.
DISCUSSION
The discussion should start with the authors evaluation of their first main research question: evaluate the performance of GF-1 high spatial resolution remote sensing imagery to estimate CC. Although the comparison of the accuracy for the three statistical algorithms is important in this paper, the most important objective proposed by authors was to evaluate the performance of GR-1 to study the CC. Thus, authors must pass the last paragraph of the discussion to the beginning of the discussion.
Reviewer 2 Report
I would say that the topic of the manuscript is a classical one in remote sensing, it is interesting and also interesting is the use and evaluation of the GF-1 satellite. The authors used classical multivariate data analysis and also more new and interesting like GAM, GLM and random forest.
However, I think that what it is missing is that the authors did not go beyond the simple application of those techniques. For example
We never saw how it is the relationship (scatter plot) between the dependent variable and the original channels or the vegetation indices used? This is missing.
We never saw for example a deep evaluation of the regression, e.g. scatter plots between the residuals and the predicted values.
What is the meaning of the developed models? For example, the models presented in Table 3, what is the physical meaning of such a model. I would expect a more physically based modeling approach.
The scatter plots in figure 2 are problematic (dots have moved).
Round 2
Reviewer 2 Report
none